# Gender Differences with Dose–Response Relationship between Serum Selenium Levels and Metabolic Syndrome—A Case-Control Study

**DOI:** 10.3390/nu11020477

**Published:** 2019-02-24

**Authors:** Chia-Wen Lu, Hao-Hsiang Chang, Kuen-Cheh Yang, Chien-Hsieh Chiang, Chien-An Yao, Kuo-Chin Huang

**Affiliations:** 1Department of Family Medicine, National Taiwan University Hospital, Taipei 10002, Taiwan; biopsycosocial@gmail.com (C.-W.L.); allanchanghs@gmail.com (H.-H.C.); quintino.yang@gmail.com (K.-C.Y.); jiansie@ntu.edu.tw (C.-H.C.); yao6638@gmail.com (C.-A.Y.); 2Department of Family Medicine, College of Medicine, National Taiwan University, Taipei 10051, Taiwan; 3Department of Family Medicine, National Taiwan University Hospital Bei-Hu Branch, Taipei 10800, Taiwan

**Keywords:** selenium, metabolic syndrome, obesity, insulin resistance, lipid

## Abstract

Few studies have investigated the association between selenium and metabolic syndrome. This study aimed to explore the associations between the serum selenium level and metabolic syndrome as well as examining each metabolic factor. In this case-control study, the participants were 1165 adults aged ≥40 (65.8 ± 10.0) years. Serum selenium was measured by inductively coupled plasma-mass spectrometry. The associations between serum selenium and metabolic syndrome were examined by multivariate logistic regression analyses. The least square means were computed by general linear models to compare the serum selenium levels in relation to the number of metabolic factors. The mean serum selenium concentration was 96.34 ± 25.90 μg/L, and it was positively correlated with waist circumference, systolic blood pressure, triglycerides, fasting glucose, and homeostatic model assessment insulin resistance (HOMA-IR) in women, but it was only correlated with fasting glucose and HOMA-IR in men. After adjustment, the odds ratios (ORs) of having metabolic syndrome increased with the selenium quartile groups (*p* for trend: <0.05), especially in women. The study demonstrated that the serum selenium levels were positively associated with metabolic syndrome following a non-linear dose–response trend. Selenium concentration was positively associated with insulin resistance in men and women, but it was associated with adiposity and lipid metabolism in women. The mechanism behind this warrants further confirmation.

## 1. Introduction 

Selenium (Se) is an antioxidative micronutrient that activates Se-containing proteins known as selenoproteins [1,2]. Among identified selenoproteins, glutathione peroxidase and selenoprotein P are more notable for their known functions of antioxidation and anti-inflammation [3,4]. Therefore, numerous investigations focused on the beneficial effects of Se exposure have tried to link it to cardiometabolic outcomes, with the emphasis mainly on type 2 diabetes (T2DM) [5,6,7,8,9]. Observational studies have shown a linear trend between risk of T2DM and Se exposure—both the serum Se level and dietary Se intake [10,11] but not the nail Se concentration [12]. In a meta-analysis summarizing five randomized controlled trials, a higher relative risk of T2DM in the Se-supplemented group than in the placebo group was reported [9]. When stratifying by gender, the association remained significant in men but not in women [13]. Also, the optimal range of Se exposure is narrow and may follow a non-linear, dose–response pattern [7,14]. 

Although studies focusing on Se and diabetes are flourishing, little is known about the association between Se and metabolic syndrome (MetS), and the conclusions remain controversial [15,16,17,18]. Although a few studies identified positive associations between the serum Se concentration and MetS only in women [15,16], or no gender differences [17], there was no significant association between serum Se concentration and MetS in the third National Health and Nutrition Examination Survey (NHANES) [18]. For obesity and dyslipidemia, general and central adiposity were negatively associated with Se levels in the NHANES [19]. Conversely, a high serum Se level was associated with increased total and non-high-density lipoprotein (HDL) cholesterol in cross-sectional studies [20,21]. However, in randomized controlled trials, Se supplementation was beneficial for decreasing total cholesterol and the total-HDL cholesterol ratio [22], or there was no significant effect between the Se-supplemented group and the placebo group [23].

Metabolic syndrome is a mixed and composite index for cardiometabolic outcomes, implying that the association between MetS and Se is complicated but deserves more detailed investigation. Therefore, we conducted this study to examine the relationship between serum Se level and MetS as well as each metabolic factor. Also, the study aimed to find a correlation between obesity, insulin resistance, and gender.

## 2. Materials and Methods

### 2.1. Study Subjects

We conducted a case control study to compare the serum Se levels between patients with and without MetS from 2007 to 2017 at the National Taiwan University Hospital. Patients who came to the outpatient department with diabetes, hypertension, hyperlipidemia, or other chronic diseases and were capable of understanding and signing the informed consent sheet were invited. A total of 1165 ambulatory males or females, aged more than 40 years, were enrolled in our study. Information about age, gender, smoking, alcohol consumption, physical activity, current medications, and previous diseases was obtained by individual interviews through questionnaires. Current smokers were defined as those smoking for more than 6 months prior to this study. Former smokers were defined as those who had not smoked for more than 12 months. Former smokers and non-smokers were grouped together as non-current smokers. Also, current alcohol drinkers were defined as those drinking more than 1 ounce of alcohol per week in the 6 months prior to this study. Former drinkers were defined as those who had quit alcohol for more than 12 months. Former drinkers and teetotalers were grouped together as non-current drinkers. Physical activity was recorded as regular exercise or not. Weight, height, systolic blood pressure (BP), and diastolic BP were measured respectively by a standard electronic scale of stadiometer and sphygmomanometer. Waist circumference (WC) was measured by a trained operator. Diabetes, hypertension, and hyperlipidemia were defined based on a self-reported history or current medication being used for those conditions. This study was approved by the Ethics Committee of National Taiwan University Hospital (201511039RINA), and written informed consent was obtained from all participants.

### 2.2. Definition of Metabolic Syndrome

Participants were considered to have MetS if they met three or more of the following criteria: WC ≥ 90 cm in men or ≥80 cm in women; serum triglycerides (TGs) ≥1.69 mmol/L; HDL cholesterol <1.03 mmol/L in men or <1.29 mmol/L in women; systolic BP ≥130 and/or diastolic BP ≥85 mmHg; and fasting glucose ≥5.56 mmol/L. Participants with medications for diabetes, hypertension, or hyperlipidemia were sorted into the group that met the criteria for fasting glucose ≥5.56 mmol/L, BP ≥130/85 mmHg, or serum TG ≥1.69 mmol/L, respectively.

### 2.3. Blood Analysis 

Venous blood samples were taken after a minimum eight-hour fasting period. Serum glucose, total cholesterol, HDL cholesterol, low-density lipoprotein (LDL) cholesterol, and TG were assessed by an automatic spectrophotometric assay (HITACHI 7250, Denka Seiken Co, Niigata, Japan). Fasting insulin level was measured by a microparticle enzyme immunoassay using an AxSYM system (Abbott Laboratories, Dainabot Co, Tokyo, Japan). The homeostatic model assessment insulin resistance (HOMA-IR) was applied as an indirect measure of the degree of insulin resistance (HOMA-IR = fasting insulin × fasting plasma glucose/22.5, with glucose in mmol/L and insulin in mU/L) [24]. Serum Se was measured using inductively coupled plasma mass spectroscopy. Serum samples were diluted 1:24 with diluents of 0.1% nitric acid and 0.1% Triton X-100. The calibration standards were prepared in a blank matrix and run using the standard addition calibration type. The serum samples were analyzed in the peak-jumping mode for ^82^Se, with the detection limit set at 0.01 μmol/L. Accuracy of the analysis was checked against Seronorm Trace Element Human Serum (batch 704121; Nycomed AS, Oslo, Norway) as reference material [6].

### 2.4. Statistical Analysis

Participants were divided into quartiles according to the serum Se levels. Data are presented as means (SDs) for continuous variables and numbers (percentage) for categorical variables. Multiple logistic regression analyses were performed to estimate the odds of having MetS among the quartiles of Se after adjusting for age, gender, current smoking status, current drinking status, physical activity, body mass index (BMI), and HOMA-IR. Tests for trends across serum Se quartiles were calculated by entering the quartile as an ordinal number in a regression model. Multiple linear regression models with each metabolic factor as dependent variables and serum Se as an independent variable were applied. Log transformation of the variables was performed if they were not normally distributed as assessed by the Kolmogorov–Smirnov test. The least square means were computed by general linear models adjusted for age, gender, current smoking status, current drinking status, and physical activity to compare serum Se concentration to the number of metabolic factors. Statistical analyses were performed using SPSS statistical software (V.17, SPSS, Chicago, IL, USA). A *p* value of <0.05 was considered to be statistically significant.

## 3. Results

The basic characteristics of the participants are shown in Table 1. The average age of the participants was 65.8 ± 10.0 years, and 64.1% were female. The mean serum Se concentration was 96.34 ± 25.90 μg/L, and the interquartile cut-off values of Se were 76.0, 94.0, and 113.7 μg/L. The serum Se levels in MetS and non-MetS groups were 102.93 ± 26.46 µg/L and 85.88 ± 21.26 µg/L, respectively. The associations of serum Se levels and prevalence of MetS by multiple logistic regression analyses are shown in Table 2. In model 1, the results showed that a higher serum Se level was correlated with a higher risk of MetS. The odds ratios (ORs) of having MetS in the second, third, and fourth Se quartile groups were 1.41 (95% CI 1.01–1.95), 2.57 (95% CI 1.83–3.59), and 5.47 (95% CI 3.75–7.96), respectively, compared with the first quartile group of serum Se level (*p* for trend: <0.001). In model 2, the results showed that a higher serum Se level was correlated with a higher risk of MetS after adjusting for age, gender, current smoking status, current drinking status, and physical activity. The ORs of having MetS in the second, third, and fourth Se quartile groups were 1.42 (95% CI 1.02–1.98), 2.39 (95% CI 1.69–3.37), and 4.96 (95% CI 3.39–7.28), respectively, compared with the first quartile (*p* for trend: <0.001). In model 3, after further adjusting for BMI, the ORs of risk for MetS in the second, third, and fourth Se quartile groups were to 1.18 (95% CI 0.80–1.73), 1.98 (95% CI 1.33–2.96), and 3.93 (95% CI 2.54–6.09), respectively, compared with the first quartile (*p* for trend: <0.001). In model 4, after further adjusting for HOMA-IR, the ORs of having MetS in the second, third, and fourth Se quartile groups decreased to 0.82 (95% CI 0.52–1.30), 1.69 (95% CI 1.03–2.79), and 1.66 (95% CI 0.88–3.12), respectively, compared with the first quartile (*p* for trend: <0.022). The interaction between Se groups and HOMA-IR was not significant (*p* = 0.057). The serum Se concentration was positively associated with WC, systolic BP, natural logarithm of TG (lnTG), fasting glucose, and HOMA-IR using multivariate linear regression analyses after adjusting for age, gender, current smoking status, current drinking status, exercise, and BMI (see Table 3).

After stratifying by gender, there was a similar higher crude OR of having MetS across the quartile groups of Se level in men (Q2: 1.89, 95% CI: 1.01–3.54; Q3: 2.32, 95% CI: 1.31–4.13; Q4: 3.63, 95% CI: 1.99–6.64, *p* for trend: <0.001) and in women (Q2: 1.26, 95% CI: 0.86–1.86; Q3: 2.57, 95%: 1.68–3.92; Q4: 7.00, 95% CI: 4.26–11.50, *p* for trend: <0.001). After adjusting for age, current smoking status, current drinking status, physical activity, and BMI, the significant trend of having a higher risk of MetS was decreased but was persistently noted more in women (Q2: 1.03, 95% CI: 0.64–1.65; Q3: 2.10, 95% CI: 1.25–3.52; Q4: 5.33, 95% CI: 2.94–9.66, *p* for trend: <0.001) than in men (Q2: 1.62, 95% CI: 0.79–3.31; Q3: 1.94, 95% CI: 0.99–3.82; Q4: 2.38, 95% CI: 1.18–4.83; *p* for trend: 0.015) (Table 4 and Figure 1). For each metabolic factor, there was a positive association with WC, systolic BP, lnTG, fasting glucose, and HOMA-IR in women, but there were only positive associations with fasting glucose and HOMA-IR in men after adjusting for age, current smoking status, current drinking status, exercise, and BMI (Table 5 and Figure 1).

The least square means (± SDs) of serum Se concentration in relation to the number of metabolic factors are shown in Figure 2A. In the linear multiple regression models, after adjusting for age, gender, current smoking status, current drinking status, and physical activity, the serum Se concentration increased with the escalation of the number of metabolic factors (test for trend: *p* < 0.001). After stratifying by gender, the serum Se concentration increased as the number of metabolic factors increased both in female and male patients after adjustment (test for trend: *p* < 0.001) (Figure 2B,C).

## 4. Discussion

The results of the present study showed a positive association between serum Se level and the risk of MetS. Also, the serum Se concentration was positively associated with WC, systolic BP, lnTG, fasting glucose, HOMA-IR, and the number of metabolic factors (test for trend: *p* < 0.001), following a dose–response relationship. Further, there was a 3.93-fold risk of MetS in the highest Se quartile compared with the lowest quartile after adjusting for demographic confounders and BMI (5.33-fold in women and 2.38-fold in men). Although further adjustment for HOMA-IR diminished most of the magnitude of the association between Se and MetS, there was a non-linear, dose–response trend whereby the odds of having MetS with the escalation of Se level (*p* for trend 0.022). These findings support a positive association between serum Se gradients and MetS independent of obesity and insulin resistance. The persistence of a direct relationship between Se exposure and risk of MetS after adjusting for BMI and HOMA-IR also implied that as-yet-unidentified confounding variables affected this association. Stratifying by gender, Se level was positively associated with insulin resistance (fasting glucose and HOMA-IR) in men and women, but with adiposity and lipid metabolism (WC, SBP, and lnTG) in women only, implying an effect modification by dimorphic gender. 

The overall findings of this study are in agreement with the majority of previous observational studies, which reported positive associations between serum Se level and MetS [15,16,17]. In a Chinese case-control study, a higher level of plasma Se was associated with an increased risk of MetS both in men and women [15]. Similarly, the IMMIDIET (The dietary habit profile in European communities with different risk of myocardial infarction: the impact of migration as a model of gene-environment interaction) project and an observational study in Lebanon showed a positive association between serum Se and MetS, but only in women [16,17]. Conversely, there was no significant association between serum Se and MetS in the third NHANES [18]. In animal models, knockout mice under adequate Se diets developed MetS pattern including hyperinsulinemia, increased body weight, dyslipidemia, and glucose intolerance [25]. The potential mechanism to link Se to insulin resistance and obesity may be partly mediated by glutathione peroxidase and selenoprotein P, due to their notable anti-inflammation functions [7]. Associated with the attenuation of antioxidative actions, Se-supplemented rats were found to develop insulin resistance [26]. Also, the overexpression of glutathione peroxidase induced the development of insulin resistance and obesity in mice [27]. Furthermore, there was an association between gene polymorphism of selenoprotein P and fasting insulin in a human study, supporting the role of selenoprotein P in glucose metabolism [28]. In terms of gender differences, glutathione peroxidase overexpression with hyperinsulinemia was only observed in male mice [29], whereas the expression of glutathione peroxidase in liver was observed more in female-derived cells compared to male-derived cells [30] Moreover, elevated selenoprotein P and insulin resistance were only observed in female mice [31]. In terms of human gene investigations, there was an elevated expression of glutathione peroxidase and selenoprotein P genes in women in relation to obesity in the England SELGEN study [32], while glutathione peroxidase polymorphisms were related to an increased incidence of MetS in men in a Japanese adult cohort [33]. In a Finnish cohort, variation in the selenoprotein S gene locus was associated with coronary heart disease and ischemic stroke in women [34]. There were gender differences in the amount of Se necessary to reach optimal Se expression.

A meta-analysis that pooled five observational studies of 13,460 subjects found that there was a positive, non-linear, dose–response association between serum Se levels and T2DM [14]. Also, there was a higher relative risk of T2DM in the Se-supplemented group than the placebo group in a meta-analysis summarizing five randomized controlled trials [9]. In animal studies, both overexpression and deficiency of selenoproteins can promote the development of T2DM, following a non-linear correlation [35]. For adiposity, serum Se was inversely associated with BMI in both men and women, whereas it was associated with the percentage of body fat only in women in the third NHANES [19]. However, there were consistent and positive associations between the Se concentration and total cholesterol [21,36], TG [37], and non-HDL cholesterol [21]. These outcomes were also identified in the UK PRECISE (Prevention of Cancer by Intervention with Selenium) study [22] but not in a study of an elderly Danish population’s [23] risk of elevated lipid profiles in the Se-supplementation group. Little is known about the relationship between Se and hypertension. Although some studies have showed positive associations between Se and systolic and diastolic BP [21,34], there was no association shown between Se and hypertension in a systemic review [38]. Generally speaking, previous observational and randomized controlled studies have elucidated a positive trend between Se and MetS, but the differences between genders are still debated. This might be related to unequal organ distribution [30], different optimal levels for Se expression [7,29,31], and polymorphisms in different genders [32,33,34]. In our study, we further confirmed the dose–response association between Se and MetS and found that dimorphic genders differed in response to insulin resistance, adiposity, and lipid profiles in relation to the Se level. 

There are some limitations to our study. First, we were not able to establish the causal relationship between serum Se concentration and MetS because of the cross-sectional design. Although we collected and adjusted for probable confounders in our study, there could be unmeasured and undefined factors with possible residual effects. For example, there were potential influences of the duration of cardiometabolic diseases on lowering the serum Se level over time, but we did not measure the time elapsed during the development of metabolic factors among individuals with or without MetS. Moreover, the serum Se level could be altered by dietary sources of Se, including soybeans, bamboo shoots, broccoli, mushrooms, cereals, Brazil nuts, and milk powder [39]. Because we did not record the daily micronutrient supplementation and personal eating habits, there might be bias independent of MetS. We checked the total serum Se to represent the serum selenoprotein concentration and activity, but we did not determine the proportion of other forms of Se and their activities. Furthermore, we used HOMA-IR as an indirect approach to estimate the degree of insulin resistance instead of accurate dynamic techniques, such as using a euglycemic clamp. Nonetheless, this is the first human study to comprehensively demonstrate the dose–response relationship between Se and MetS and metabolic factors with a large sample size. Gender stratification analyses clearly highlighted the gender differences in insulin resistance, adiposity, and lipid metabolism. However, the underlying mechanisms need further investigation. 

## Figures and Tables

**Figure 1 nutrients-11-00477-f001:**
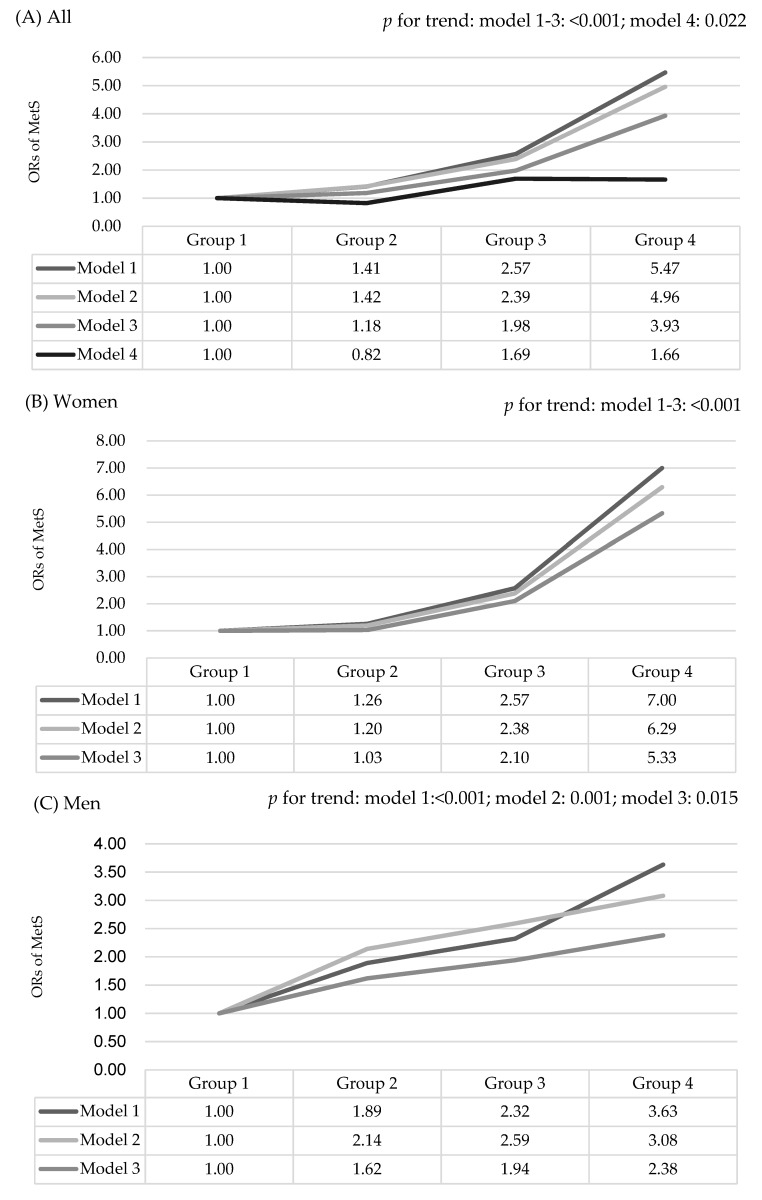
Nonlinear dose–response relationship between selenium and metabolic syndrome. (**A**) All subjects; (**B**) Female subjects; (**C**) Male subjects.

**Figure 2 nutrients-11-00477-f002:**
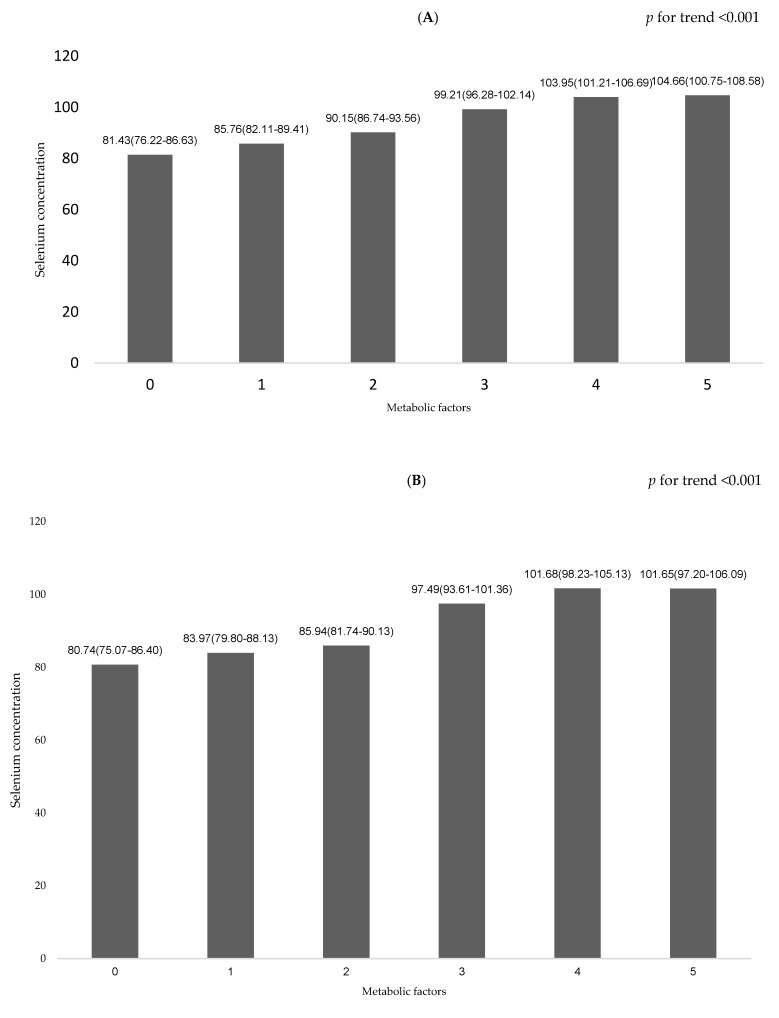
Comparison of serum selenium concentration in relation to number of metabolic factors. **(A)** All subjects; **(B)** Female subjects; **(C)** Male subjects.

**Table 1 nutrients-11-00477-t001:** Characteristics of the study population by quartiles of serum selenium levels.

	Quartiles of Serum Selenium Levels			
	Q1 (*n* = 292) (≤76.0 μg/L)	Q2 (*n* = 290) (76.1–94.0 μg/L)	Q3 (*n* = 292) (94.1–113.7 μg/L)	Q4 (*n* = 291) (>113.7 μg/L)
Gender				
Female (%)	208 (71.2)	208 (71.7)	167 (57.2)	164 (56.4)
Male (%)	84 (28.8)	82 (28.3)	125 (42.8)	127 (43.6)
Age (years)	65.8 ± 10.3	65.9 ± 9.7	66.7 ± 9.6	64.9 ± 10.3
BMI (kg/m^2^)	24.1 ± 3.5	24.8 ± 4.1	25.5 ± 4.3	26.5 ± 4.5
WC (cm)	82.9 ± 9.3	85.3 ± 10.6	87.6 ± 11.0	90.8 ± 11.1
Systolic BP	127.1 ± 16.8	128.0 ± 14.7	131.4 ± 15.6	159.6 ± 9.0
Diastolic BP	75.6 ± 11.0	76.2 ± 9.2	76.2 ± 10.1	68.0 ± 14.9
TCHO (mmol/L)	5.28 ± 0.95	5.05 ± 1.04	4.90 ± 1.03	4.59 ± 0.98
TGs (mmol/L)	1.51 ± 0.93	1.58 ± 1.29	1.59 ± 0.81	1.77 ± 1.23
HDL-C (mmol/L)	1.35 ± 0.32	1.33 ± 0.31	1.29 ± 0.35	1.25 ± 0.33
LDL-C (mmol/L)	3.20 ± 0.74	3.00 ± 0.80	2.94 ± 0.81	2.68 ± 0.80
Glu (mmol/L)	5.89 ± 1.47	6.22 ± 1.66	6.76 ± 2.06	7.23 ± 2.21
Insulin (U/mL)	8.30 ± 5.86	9.35 ± 7.71	10.68 ± 8.02	13.12 ± 8.71
HOMA-IR	2.28 ± 2.18	2.49 ± 2.74	3.09 ± 2.83	3.49 ± 3.07
Selenium (µg/L)	65.13 ± 7.81	85.16 ± 5.19	104.46 ± 5.59	130.66 ± 14.82
Cigarette (%)	15 (5.1)	27 (9.3)	42 (14.4)	55 (18.9)
Alcohol (%)	21 (7.2)	32 (11.0)	46 (15.8)	53 (18.2)
Exercise (%)	192 (65.8)	199 (68.6)	180 (61.6)	163 (56.0)
Diabetes (%)	76 (26.0)	114 (39.3)	183 (62.7)	247 (84.9)
Hypertension (%)	117 (40.1)	138 (47.6)	192 (65.8)	210 (75.3)
Hyperlipidemia (%)	91 (31.2)	129 (44.5)	167 (57.2)	219 (75.3)
Elevated WC (%) *	141 (48.3)	172 (59.3)	173 (59.2)	220 (75.6)
High TG (%) *	141 (48.3)	163 (56.2)	193 (66.1)	220 (75.6)
Low HDL-C (%) *	111 (38.0)	114 (39.3)	112 (38.4)	132 (45.4)
Elevated BP (%) *	134 (45.9)	139 (47.9)	183 (62.7)	186 (63.9)
IFG (%) *	136 (46.6)	170 (58.6)	213 (72.9)	256 (88.0)
Metabolic factors	2.27 ± 1.49	2.61 ± 1.48	2.99 ± 1.39	3.49 ± 1.20
MetS (%)	129 (44.2)	151 (52.1)	194 (66.4)	235 (80.8)

Abbreviations: BMI: body mass index; WC: waist circumference; BP: blood pressure; TCHO: total cholesterol; TGs: triglycerides; HDL-C: high-density lipoprotein cholesterol; LDL-C: low-density lipoprotein cholesterol; Glu: fasting glucose; HOMA-IR: homeostasis model assessment of insulin resistance; IFG: impaired fasting glucose; MetS: metabolic syndrome. * Elevated WC: WC ≥90 cm in men or ≥80 cm in women; High TG: serum TG ≥1.69 mmol/L; Low HDL-C: HDL-C <1.03 mmol/L in men or <1.29 mmol/L in women; Elevated BP: systolic BP ≥130 and/or diastolic BP ≥85 mmHg; and IFG: impaired fasting glucose ≥5.56 mmol/L. Continuous variables are presented by mean ± SD and categorical variables are presented as the percentage of participants (%).

**Table 2 nutrients-11-00477-t002:** Odds ratios (ORs) of having MetS derived from multiple logistic regression analyses in quartiles of serum selenium levels.

	Quartile of Serum Selenium Levels				
	Q1 (*n* = 292) (≤ 76.0 μg/L)	Q2 (*n* = 290) (76.1–94.0 μg/L)	Q3 (*n* = 292) (94.1–113.7 μg/L)	Q4 (*n* = 291) (>113.7 μg/L)	*p*-value of Se Tertile
MetS, *n* (%)	129 (44.2)	151 (52.1)	194 (66.4)	235 (80.8)	
Model 1	1.00	1.41 (1.01–1.95)	2.57 (1.83–3.59)	5.47 (3.75–7.96)	<0.001
Model 2	1.00	1.42 (1.02–1.98)	2.39 (1.69–3.37)	4.96 (3.39–7.28)	<0.001
Model 3	1.00	1.18 (0.80–1.73)	1.98 (1.33–2.96)	3.93 (2.54–6.09)	<0.001
Model 4	1.00	0.82 (0.52–1.30)	1.69 (1.03–2.79)	1.66 (0.88–3.12)	0.022

Model 1: No adjustment; Model 2: adjusted for age, gender, current smoking status, current drinking status, and physical activity; Model 3 adjusted for variables in model 2, plus BMI as a confounding factor. Odds ratio of BMI (95% confidence interval [CI] 1.41–1.58, *p* < 0.001); Model 4: adjusted for variables in model 3, plus HOMA-IR as a confounding factor. Odds ratio of elevated HOMA-IR (95% CI 1.94–2.90, *p* < 0.001); HOMA-IR: homeostasis model assessment of insulin resistance; MetS: metabolic syndrome.

**Table 3 nutrients-11-00477-t003:** Linear regression models showing standardized betas with serum selenium concentrations as independent variable for metabolic factors.

	WC		Systolic BP		Diastolic BP		lnTG		HDL-C		Fasting Glucose		HOMA-IR	
	Beta	*p*	Beta	*p*	Beta	*p*	Beta	*p*	Beta	*p*	Beta	*p*	Beta	*p*
Model 1	0.260	<0.001	0.159	<0.001	0.052	0.076	0.127	<0.001	–0.070	0.021	0.252	<0.001	0.172	<0.001
Model 2	0.284	<0.001	0.118	<0.001	0.029	0.313	0.075	0.010	–0.002	0.940	0.210	<0.001	0.135	0.001
Model 3	0.231	<0.001	0.119	<0.001	0.026	0.363	0.067	0.022	0.005	0.873	0.204	<0.001	0.132	0.001

Abbreviations: WC: waist circumference; BP: blood pressure; lnTG: natural logarithm of TG; HDL-C: high-density lipoprotein cholesterol; HOMA-IR: homeostasis model assessment of insulin resistance. Model 1: adjusted for age, gender; Model 2: adjusted for age, gender, current smoking status, current drinking status, and physical activity; Model 3: adjusted for age, gender, current smoking status, current drinking status, physical activity, and BMI.

**Table 4 nutrients-11-00477-t004:** Odds ratios (ORs) of having MetS derived from multiple logistic regression analyses in quartiles of serum selenium levels, stratified by gender.

	Quartile of Serum Selenium Levels				
	Q1 (*n* = 292) (≤76.0)	Q2 (*n* = 290) (76.1–94.0)	Q3 (*n* = 292) (94.1–113.7)	Q4 (*n* = 291) (>113.7)	*p*-value of Se
Female					
MetS, *n* (%)	87/207 (40.2)	107/205 (47.8)	108/166 (65.1)	137/164 (83.5)	
Model 1	1.00	1.26 (0.86–1.86)	2.57 (1.68–3.92)	7.00 (4.26–11.50)	<0.001
Model 2	1.00	1.20 (0.81–1.80)	2.38 (1.55–3.66)	6.29 (3.78–10.45)	<0.001
Model 3	1.00	1.03 (0.64–1.65)	2.10 (1.25–3.52)	5.33 (2.94–9.66)	<0.001
Male					
MetS, *n* (%)	42/84 (50)	53/81 (65.4)	86/123 (69.9)	98/125 (78.4)	
Model 1	1.00	1.89 (1.01–3.54)	2.32 (1.31–4.13)	3.63 (1.99–6.64)	<0.001
Model 2	1.00	2.14 (1.10–4.15)	2.59 (1.40–4.79)	3.08 (1.63–5.83)	0.001
Model 3	1.00	1.62 (0.79–3.31)	1.94 (0.99–3.82)	2.38 (1.18–4.83)	0.015

Model 1: adjusted for age; Model 2: adjusted for age, current smoking status, current drinking status, and physical activity; Model 3 adjusted for variables in model 2, plus BMI as a confounding factor. Odds ratio of BMI (95% confidence interval [CI] 1.44–1.68, *p* < 0.001 for male; 95% CI 1.26–1.51, *p* < 0.001 for female). MetS: metabolic syndrome.

**Table 5 nutrients-11-00477-t005:** Linear regression models showing standardized betas with serum selenium concentrations as independent variable for metabolic factors, stratified by gender.

	WC		Systolic BP		Diastolic BP		lnTG		HDL-C		Fasting Glucose		HOMA-IR	
	Beta	*p*	Beta	*p*	Beta	*p*	Beta	*p*	Beta	*p*	Beta	*p*	Beta	*p*
Female														
Model 1	0.266	<0.001	0.197	<0.001	0.077	0.035	0.184	<0.001	–0.067	<0.001	0.271	<0.001	0.192	<0.001
Model 2	0.234	<0.001	0.184	<0.001	0.095	0.010	0.164	<0.001	–0.049	0.186	0.219	<0.001	0.166	<0.001
Model 3	0.056	0.002	0.139	<0.001	0.069	0.064	0.108	0.003	0.012	0.738	0.162	<0.001	0.083	0.035
Male														
Model 1	0.228	<0.001	0.072	0.141	0.022	0.450	0.0.39	0.430	–0.078	0.111	0.230	<0.001	0.129	0.048
Model 2	0.168	0.001	0.076	0.142	–0.028	0.580	-0.027	0.589	–0.072	0.155	0.234	<0.001	0.171	0.009
Model 3	0.048	0.056	0.051	0.321	–0.048	0.349	-0.052	0.295	–0.010	0.843	0.211	<0.001	0.158	0.008

Abbreviations: WC: waist circumference; BP: blood pressure; HDL-C: high-density lipoprotein cholesterol; HOMA-IR: homeostasis model assessment of insulin resistance. Model 1: adjusted for age; Model 2: adjusted for age, current smoking status, current drinking status, and physical activity; Model 3: adjusted for age, current smoking status, current drinking status, physical activity, and BMI.

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
