# Peer review of "Gender Differences with Dose–Response Relationship between Serum Selenium Levels and Metabolic Syndrome—A Case-Control Study"

_nutrients, 2019, doi:10.3390/nu11020477_

Round 1
Reviewer 1 Report
This study explored the associations between serum selenium (Se) levels and metabolic syndrome (MetS) and its various metabolic factors in 1,165 out-patients aged ≥40 (65.8±10.0) years. The mean serum selenium concentration was 96.34±25.90 μg/L and was positively correlated with components of MetS, such as waist circumference, systolic blood pressure, triglycerides, fasting glucose, and HOMA-IR in women, but only with fasting glucose and HOMA-IR in men. Though the study did not provide any causality, it is the first human study to demonstrate a dose-response relationship between Se levels and MetS in a large number of patients. I have some concerns:
1. P9 L196: “These findings supported that obesity and insulin resistance were either a cause or a consequence between serum Se gradients and MetS.“ This is a confusing statement, as patients with insulin resistance are generally overweight or obese, while not all obese individuals have insulin resistance. It should therefore be clarified whether Se, by increasing BW, may influence the development of insulin resistance. It is difficult to explain how obesity could increase Se status.
2. There is no mention of the mechanisms of Se involvement in obesity and insulin resistance.
3. What might be the mechanism behind the association of serum Se with hyperchosterolemia in women but not in men?
Minor points
P9 L214: “GP polymorphisms were in related to…”delete in
P9 L 217 “Digging detail into each metabolic factor, there was a non-linear…”
Please rephrase as the sentence is not clear.
The paper needs a thorough linguistic revision by an English native speaker.
Author Response
A point-by-point response to the reviewer’s comments has uploaded as a Word file.

Reviewer 2 Report
Reviewer’s comment
This article by Dr. Lu and colleagues primarily focuses on the relationship between serum Se levels and Met S. It seems to be interesting that there was a gender difference between the relationship. However, several revisions are required for the better quality of the issue.
Major
#1. Insulin resistance is defined as the value of HOMA-IR exceeding 2.5. However, the mean value in each quartile is less than 2.5, implying that most of the participants did not have insulin resistance. Therefore, the prevalence of insulin resistance in the participants should be elucidated.
#2. The authors should assign the group of normal healthy control (NHC) and compare the serum Se levels between the groups of Met S and NHC.
#3. The relationship between serum Se level and the number of metabolic factor should be analyzed in male and female group, respectively (Figure 2).
#4. The authors should mention the reasons for a gender difference between the serum Se level and Met S in “Discussion”. It is likely to become the most important point in this study.
#5. A previous study revealed that lower and higher serum Se levels resulted in higher prevalence of type 2 DM (Wang et al, Nutr J, 2016). However, only higher Se levels were responsible for Met S in this study. The authors should describe the dissociations between the previous study and this study.
Minor
#1. GP should be spell out to glutathione peroxidase.
#2. Table 4B should be replaced with Table 3B.
#3. The page numbers are missing in references 7 and 19.
#4. “Faseb J” should be corrected to “FASEB J” in reference 28.
Author Response

(The authors gave the same response as above.)

Round 2
Reviewer 1 Report
No further comments
Author Response
Thanks for your expertise, support and feedback.
Reviewer 2 Report
The quality of this revised article seemed to be better than that of original one.
#1. The authors compared serum Se levels between the groups of Met S and non-Met S. They did not describe the assigned patients with non-Met S in detail. Patients with non-Met S may include those who had only insulin resistance in this study. The authors should compare serum Se levels between the groups of patients with Met S and normal health controls.
#2. The authors described that serum Se level were associated with only insulin resistance in men, indicating that serum Se levels did not affect hypertension, dyslipidemia, and waist circumference (Table 3). But, as the numbers of metabolic factors were increased, serum Se levels were gradually elevated in Figure 2C. These results seemed to be paradoxical.
Author Response
Thanks for your expertise and feedback. We answered your comments point-by-point in attached files.
